# Microbiota succession influences nematode physiology in a beetle microcosm ecosystem

Wen-Sui Lo[1,2], Ralf J. Sommer ✉[2] & Ziduan Han ✉[2,3]

Unravelling the multifaceted and bidirectional interactions between microbiota and host physiology represents a major scientific challenge. Here, we utilise the nematode model, *Pristionchus pacificus*, coupled to a laboratory-simulated decay process of its insect host, to mimic natural microbiota succession and investigate associated tripartite interactions. Metagenomics reveal that during initial decay stages, the population of vitamin B-producing bacteria diminishes, potentially due to a preferential selection by nematodes. As decay progresses to nutrient-depleted stages, bacteria with smaller genomes producing less nutrients become more prevalent. Lipid utilisation and dauer formation, representing key nematode survival strategies, are influenced by microbiota changes. Additionally, horizontally acquired cellulases extend the nematodes' reproductive phase due to more efficient foraging. Lastly, the expressions of *Pristionchus* species-specific genes are more responsive to natural microbiota compared to conserved genes, suggesting their importance in the organisms' adaptation to its ecological niche. In summary, we show the importance of microbial successions and their reciprocal interaction with nematodes for insect decay in semi-artificial ecosystems.

Among various environmental stimuli, microbiota play a crucial role in regulating the physiology of animals[1]. Bacterial-feeding nematodes are powerful organisms to study host–microbiota interactions due to their simple anatomy, rapid generation time and powerful molecular tools[2–4]. Studies, predominantly featuring the model nematode *Caenorhabditis elegans* (*C. elegans*), illustrate how singular bacterial strains can profoundly influence the host's physiology and behaviour[2,5–7]. Yet, in natural settings, nematodes interact with diverse bacterial communities that change rapidly over time and represent effects from beneficial to pathogenic, but the holistic interaction between microbiota and nematodes, as well as the bacterial response to nematode foraging, have rarely been explored.

*Pristionchus pacificus* (*P. pacificus*), an omnivorous nematode exhibiting a unique life cycle[8], transitions between survival and foraging stages in response to environmental cues. Global sampling has demonstrated that *P. pacificus* nematodes are widely distributed,

primarily associated with scarab beetles. Its natural habitat on La Réunion Island provides a unique beetle–microbiota ecosystem wherein *P. pacificus* thrives on decaying beetles[9,10], progressing from dauer, an arrested developmental stage, to active stages feeding upon beetle death, and reverting back to dauer as resources depleted[11]. Also, *P. pacificus* is a potential predator of other nematodes on the decaying insect carcass, which has resulted in a plethora of investigations studying the regulation and the ecological and evolutionary significance of associated mouth-form plasticity and self-recognition mechanisms. These characteristics make *P. pacificus* a primary model for 'eco–evo–devo' studies[12,13].

However, two aspects about the tripartite interactions between bacteria, nematodes and insects remained previously unknown. First, bacterial succession due to nematode foraging during the decay of the beetle carcass has only been investigated using 16S rDNA-based approaches, resulting in limited resolution of species identity and

[1]Institute of Future Agriculture, Northwest A&F University, Yangling, Shaanxi 712100, China. [2]Department for Integrative Evolutionary Biology, Max Planck Institute for Biology, Tübingen 72076, Germany. [3]State Key Laboratory for Crop Stress Resistance and High-Efficiency Production, College of Plant Protection, Northwest A&F University, Yangling, Shaanxi 712100, China. ✉e-mail: ralf.sommer@tuebingen.mpg.de; ziduan.han@nwafu.edu.cn

functional diversity. Second, the inherent genetic diversity of wild-collected nematodes, limited a proper understanding of the effects of the microbiota on *P. pacificus*. Consequently, how bacterial communities change over time and how *P. pacificus* responds to microbial succession are currently unclear.

In this work, to mitigate the inherent variability of natural settings, we create a laboratory microcosm to investigate the dynamics and succession of bacterial composition and the corresponding *P. pacificus* physiology. We employ farmed grubs from two beetle species

belonging to the Scarabaeidae family as a source of organic material to simulate the microbial environment found in the decomposition process of insects. This microcosm allows studying insect decay, one of the most prevalent but understudied processes in soil ecosystems, under controlled laboratory conditions. Specifically, *P. pacificus* nematodes are introduced to investigate the general effects of natural microbiota. We also recruit a *P. pacificus* CRISPR/Cas9-knockout mutant lacking the horizontally acquired cellulolytic ability which can assist *P. pacificus* to disrupt bacterial biofilms[14]. Bacteria and both

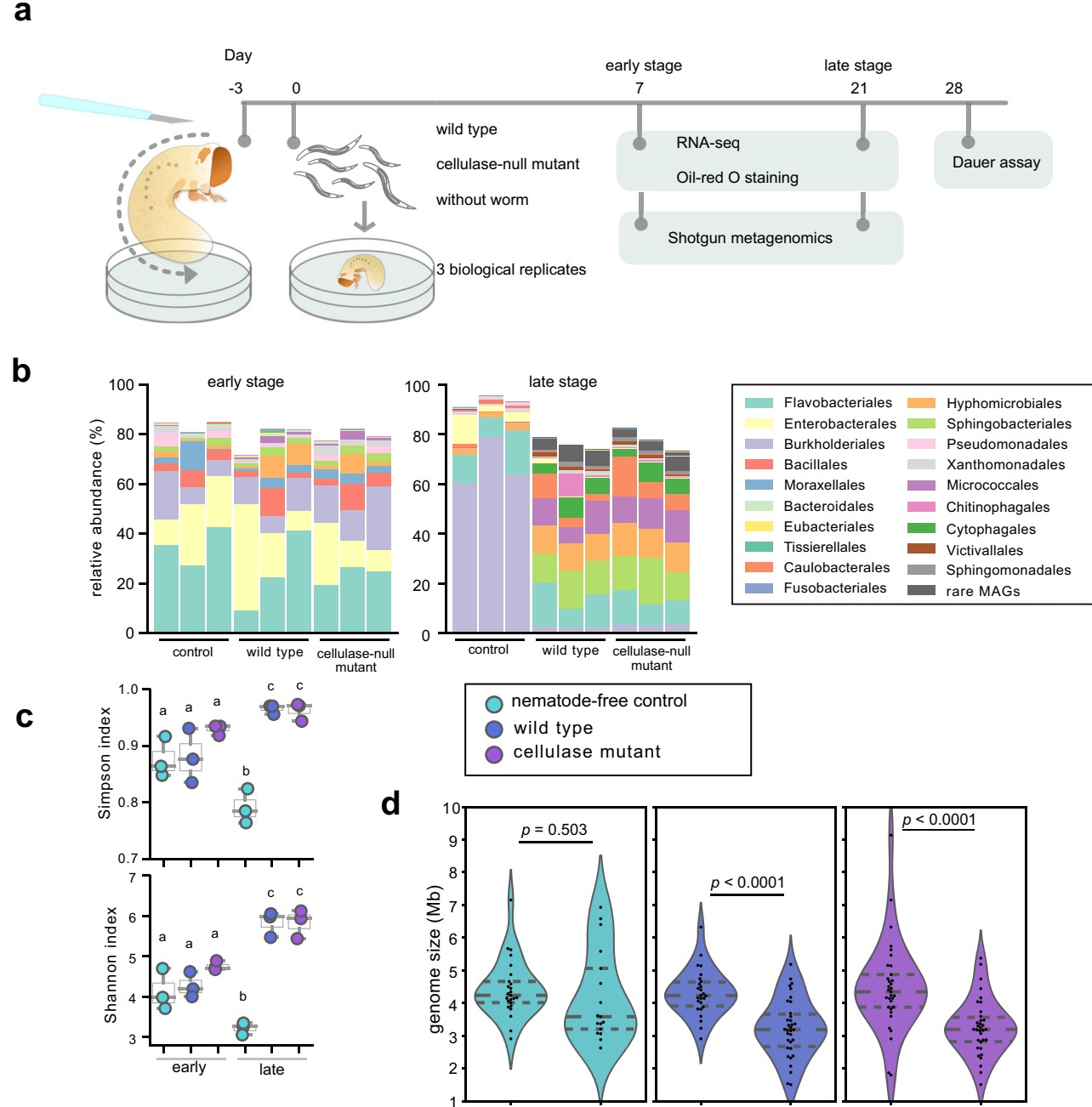

**Fig. 1 | Taxonomic composition of metagenome-assembled genomes (MAGs) on grub carcasses. a** An illustration of the experimental design. **b** Taxonomic comparison at the order level. MAGs reconstructed from early (left) and late stage (right) with ranks ordered from bottom to top by their decreasing proportion. Only the top 30 most abundant orders are shown. The remaining lineages are grouped as 'others'. **c** Alpha diversity (Simpson index) and evenness (Shannon evenness index) were measured using MAGs from three biological replicates in each treatment. The box plot's center features the median at the 50th percentile, with the $Q_1$

(25th percentile) and $Q_3$ (75th percentile) marking the lower and upper bound of the data. Whiskers extend from these quartiles to the minimum and maximum values as $Q_1 - 1.5$ IQR and $Q_3 + 1.5$ IQR, respectively. Significantly different groups are indicated by letters a, b, and c, determined by pairwise Tukey HSD tests, which are two-sided and adjust for multiple comparisons. **d** Distribution of genome sizes for MAGs at the early and late stages. Source data are provided as a Source Data file.

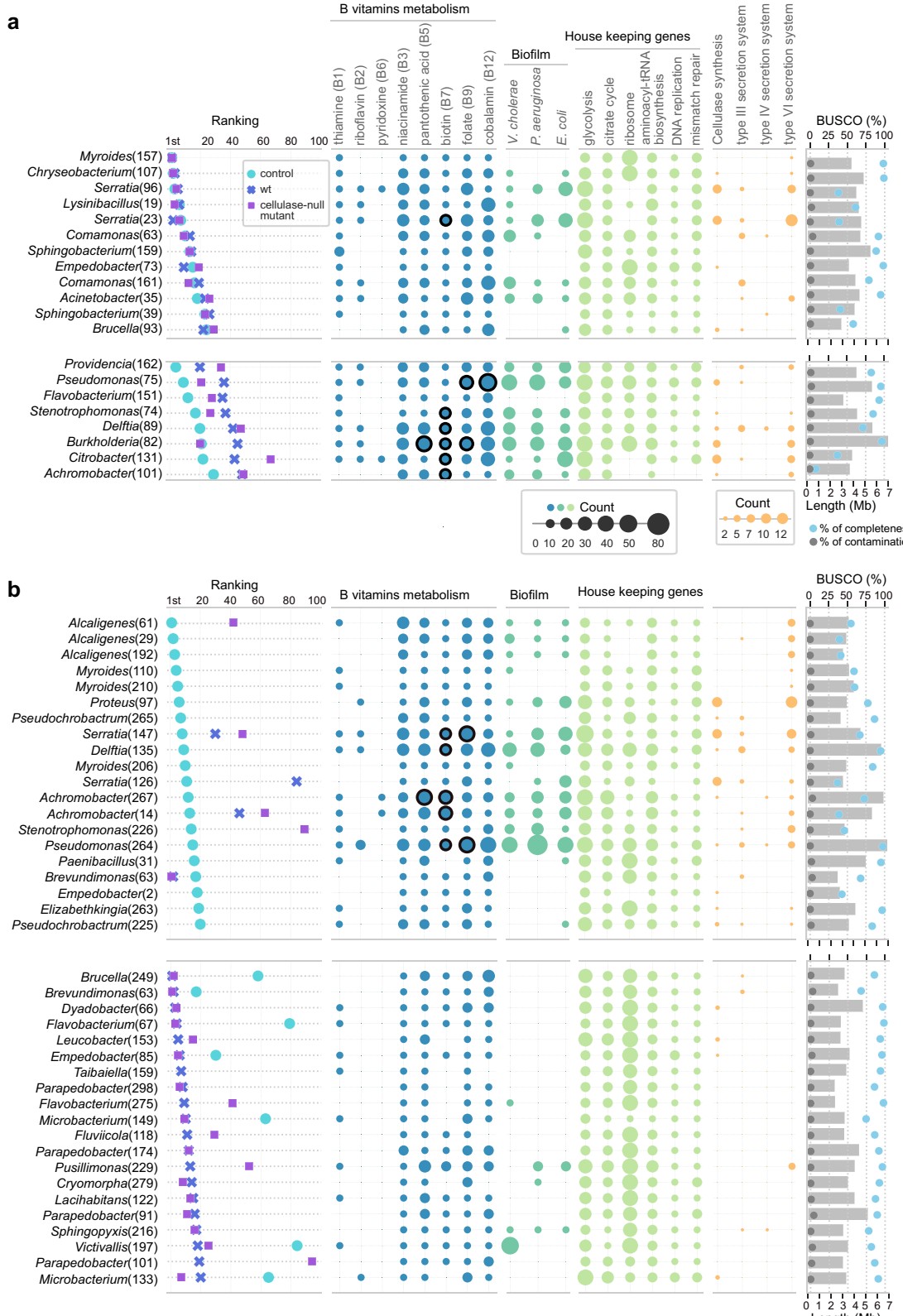

**Fig. 2 | Characteristics of metagenome-assembled genomes (MAGs) recovered from grub carcasses during early and late stages of decomposition.** The abundance in ranking, gene numbers in metabolism pathways, genome size (grey bar) and the BUSCO completeness (blue dot), and contamination (grey dot) values are shown for **a** early stage and **b** late stage. The numbers preceding the bacterial genus corresponding to the number of MAG bins are listed in Supplementary Data 1. Circles with black outlines in pantothenic acid, biotin, folate and cobalamin pathways indicate MAGs can synthesise these vitamins de novo. The remaining B vitamins pathways are relatively simple, with MAGs possessing the genes able to synthesise products de novo. **a** MAGs from the early stage do not significantly change the abundance ranking (upper), and MAGs decrease in abundance in the presence of *P*. pacificus (lower). **b** MAGs from the late stage are dominant in the control group (upper), and MAGs are dominant in the presence of the *P. pacificus* wild type (lower).

nematode strains are sampled at two points during the decomposition of rose chafer (*Cetonia aurata* (*C. aurata*)) grubs, representing nutrient-rich and nutrient-limited conditions. We utilise shotgun metagenomics to characterise bacterial population structure and genome diversity dynamics, and explore the impact of microbiota on *P. pacificus* gene expression and metabolism, focusing on fatty acid metabolism and dauer formation. We then utilise the white-spotted flower chafer (*Protaetia brevitarsis* (*P. brevitarsis*)) to confirm the microbiota composition and physiological responses in nematodes. Thus, we conduct a comprehensive study to reveal interactions between natural microbiota and *P. pacificus* in a decaying insect ecosystem.

## Results

### Metagenomic changes in decomposed grub driven by nematodes

We dissected grubs of the rose chafer *C. aurata* to mimic the natural death of scarab beetles and inoculated the carcasses of these nematode-free insects with 500 mixed-stage *P. pacificus* animals (Fig. 1a). Subsequently, we sampled bacteria and nematodes on days 7 and 21 after the nematode inoculation, representing a nutrient-rich and nutrient-limited condition (referred to as 'early stage' and 'late stage', respectively; Fig. 1a). In this study, we analysed ~327 million and 313 million read pairs from the early and late stages, respectively. After de novo assembly and subsequent removal of lower quality metagenome-assembled genomes (MAGs), we retrieved 86 MAGs for the early stage and 199 for the late stage (Supplementary Data 1). Importantly, 76% of the reads from the early stage and 63% from the late stage were mapped back to these respective MAGs. The relative abundance of MAGs was determined by mapping raw sequence reads back to the assembled contigs, followed by calculating the coverage of each MAG. Detailed information of each MAG, including genomic features, predicted species, relative abundance, and read statistics, is provided in Supplementary Data 1 and Supplementary Fig 1.

We observed major differences in the microbiota structure between the early and late stages (Fig. 1b). Surprisingly, at the early stage of the succession, the microbiota structures of the three treatments were similar, dominated by the orders Flavobacteriales, Enterobacterales, and Burkholderiales (Fig. 1b, c; Supplementary Fig. 2a). In contrast, at the late stage, the nematode-free control samples were dominated by the Burkholderiales, whereas bacterial species diversity was significantly increased in the nematode-containing samples (Fig. 1b, c; Supplementary Fig. 2a). These communities at the late stage were also largely different from those observed at the early stage, and they were dominated by the orders Flavobacteriales, Sphingobacteriales, Hyphomicrobiales, and Micrococcales (Fig. 2b). *Pristionchus* nematodes have acquired cellulase genes through horizontal gene transfer from a eukaryotic donor, and it has been shown that these cellulase proteins are secreted to the environment to digest biofilm[14]. We found that although there was no significant difference in the structure of microbiota between *P. pacificus* wild type and cellulase-null mutant at either stage (Supplementary Fig. 2a), the relative abundances of several bacterial species were changed between the two nematode strains (Fig. 2). Thus, our semi-artificial microcosm clearly shows that the presence of *P. pacificus* has a strong influence on the microbiota.

In comparison to previous studies using the rhinoceros beetle *Oryctes borbonicus* collected from natural habitats on La Réunion Island and analysed by 16S rDNA-based profiling, our results showed that the bacterial genera from the early rose chafer grub decomposition overlapped with those in wild beetle carcasses. Specifically, of the top 30 most abundant genus detected in the decomposed rose beetle grub, 26 were also detected in beetle carcasses collected in the field (Supplementary Fig. 2b). Additionally, 16S rDNA profiling of decomposed white-spotted flower chafer grubs, conducted under the same

laboratory conditions as those for the rose chafer grubs, indicates similar microbiota between these grubs. Specifically, 20 out of the 25 most abundant genera were identical to those found in the microbiota of rose chafer grubs (Supplementary Fig. 2c, d). This observation indicates that decomposing insect carcasses harbours similar microbiota compositions, and the complexity of grub carcass-associated microbiota we created is equivalent to those from nature.

### Nematodes contribute to vitamin B-producing bacteria decline

We monitored the temporal dynamics of the bacterial community by assessing the abundance of MAGs. We ranked the average percentage of the abundance from three biological replicates and identified MAGs that changed in ranking. At the early stage of decomposition, the presence of the wild type or cellulase-null mutant of *P. pacificus* did not significantly alter the abundance ranking of many MAGs (Fig. 2a). These MAGs may be bacteria less preferred or indigestible by *P. pacificus* nematodes. For instance, while *Lysinibacillus* sp. promotes *P. pacificus* growth[3], some *Serratia* spp. are nematode pathogens[15,16], illustrating a spectrum of interactions from beneficial to antagonistic.

Contrastingly, among the top 30 most abundant MAGs, the presence of *Pristionchus* nematodes significantly reduced the abundance ranking of eight taxa by more than ten positions, including species from the *Pseudomonas*, *Burkholderia*, and other diverse bacterial genera (Fig. 2a). These observations suggest that at the early stage of the grub decay, the consumption of bacteria by nematodes is not proportional to bacterial abundance, and certain bacteria are highly preferred and selected as food. Intriguingly, the capability for cobalamin (vitamin B12) or biotin (vitamin B6) biosynthesis is one of the distinct features of the preferred MAGs (Fig. 2a). Additionally, our 16S rDNA-based profiling of the bacterial composition in both grub carcasses and nematode gut revealed variations in bacterial abundance (Supplementary Fig. 2c, d). Thus, our analyses suggest a selectivity of bacterial consumption by *Pristionchus*, and nematodes are able to acquire essential vitamins, i.e. riboflavin, thiamine, pyridoxine, niacinamide, pantothenic acid, biotin, folate, and cobalamin, by feeding on these preferred microbes. These results suggest that *P. pacificus* is capable of acquiring and also likely needs pivotal cofactors from the natural microbiota.

Finally, the cellulase-null mutant showed reduced efficacy in the deduction of *Pseudomonas* sp. (75) and *Burkholderia* sp. (82), which are major sources of B vitamins. In addition, the abundances of *Klebsiella* sp., *Cohnella* sp., and *Alcaligenes* sp. were increased in carcasses with *P. pacificus* cellulase mutants (Supplementary Fig. 3a). Species of the genus *Pseudomonas*, *Burkholderia*, and *Klebsiella* are known to synthesise biofilms using cellulose as a matrix component, and pathogenicity is tightly associated with biofilm formation[17–19]. Therefore, our results suggest that the cellulolytic activity of *P. pacificus* facilitates bacterial cell lysis through biofilm degradation, and in the absence of nematode cellulase, these bacteria can flourish on the grub carcass.

### Small-genome bacteria dominate at nutrient-limited stage

At the late stage of the decomposition, the presence of *P. pacificus* nematodes had a large impact on bacterial structure and resulted in a decrease of several dominant bacterial groups (Fig. 1b). Simultaneously, the presence of *Pristionchus* caused an increase in the species diversity and relative abundance of rare bacterial groups compared to the nematode-free treatment (Fig. 1c). Surprisingly, when nematodes were present, the MAGs that dominated at the later stage had smaller genome sizes compared to those bacteria at the early stage (Figs. 1d and 2b; $p < 0.0001$). The median size of the bacterial genomes (individual bacterial species account for over 0.5% abundance) was reduced by almost 1 Mb (over 25% of the total genome size) at the late stage in both treatments containing nematodes (Fig. 1d), while it did not change in the nematode-free treatment (Fig. 1d, $p = 0.50$). These MAGs had high BUSCO scores indicating that the observed small-

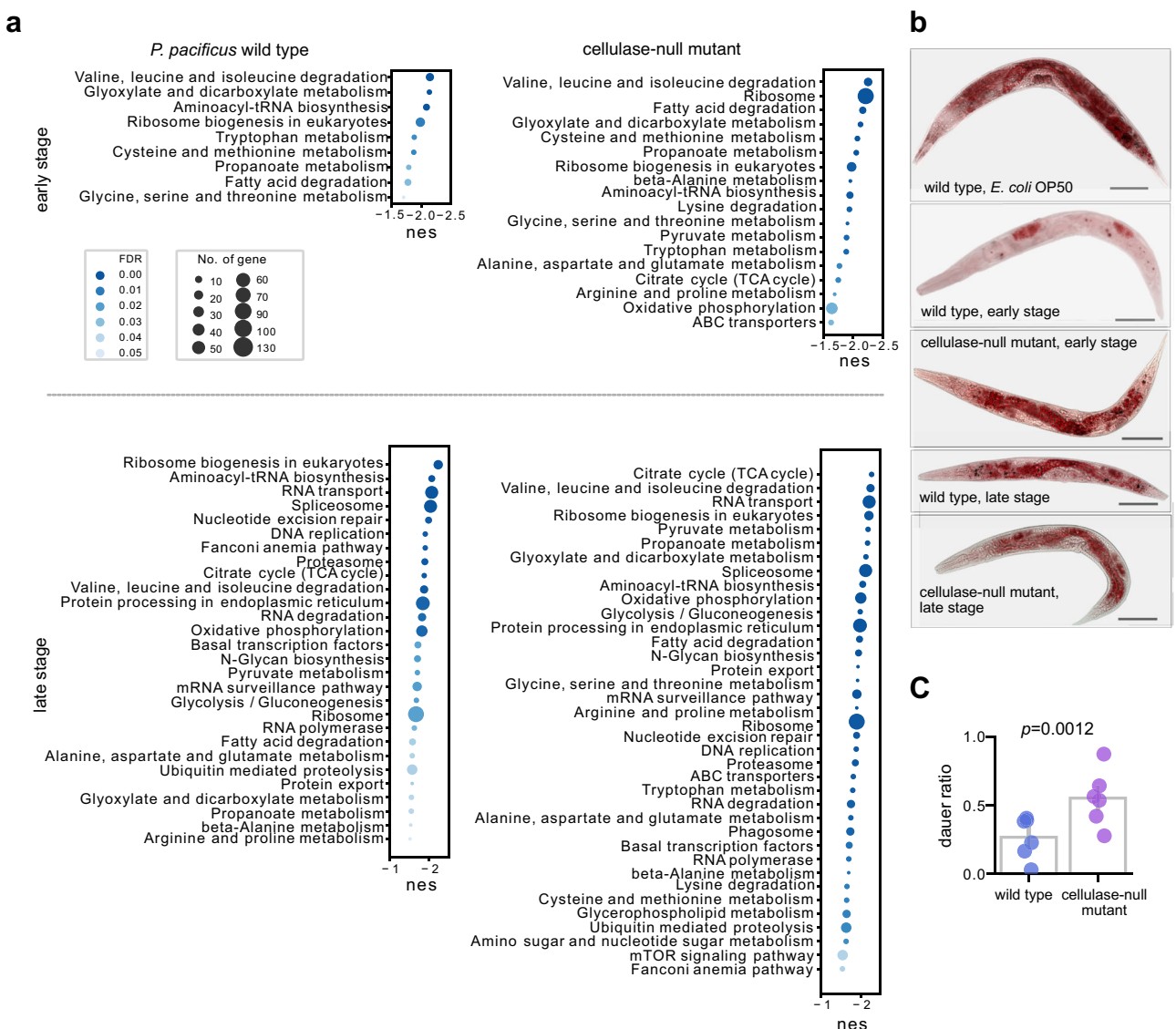

**Fig. 3 | Complex bacterial community changes the physiology of *Pristionchus pacificus*.** **a** A Gene Set Enrichment Analysis (GSEA) of *P. pacificus* metabolic pathways on grub carcasses compared to *E. coli* OP50 via the KEGG gene set. Enriched pathways with a false discovery rate adjusted p-value < 0.05 are presented. The size of the circle indicates the number of genes in the pathway. **b** Oil-red-O staining indicates lipid droplet contents on *P. pacificus*. From top to bottom, wild type on *E. coli* OP50, wild type (early stage), cellulase-null mutant (early stage), wild type (late stage), and cellulase-null mutant (late stage) on grub carcasses. Scale bar = 100 μm. **c** The proportion of dauer larvae in nematode population of *P. pacificus* wild type and cellulase-null mutant feeding on grub carcasses at the late stage. Nematodes on individual plates (*n* = 6) were counted, and a one-tailed *t*-test was performed. Source data are provided as a Source Data file.

genome sizes are unlikely due to misassemblies (Fig. 2b). The smaller genomes of these MAGs had fewer genes for cofactor synthesis pathways, as well as annotated genes for defence (e.g. biofilm and type III secretion systems) and competition (e.g. type VI and IV secretion systems) (Fig. 2b), suggesting that their dominance was not the result of resistance to nematode predation or interspecies bacterial competition.

Instead, our metagenomic assembly suggests that this dominance may be explained by different survival strategies employed by bacteria at early and late stages of decomposition. In our analyses, the early-stage MAGs distinctly lacked ribosomal protein genes, whereas these genes were found in the MAGs from the later stages of decomposition. Metagenomic binning is dependent on the tetranucleotide frequencies of contigs. Fast-growing bacteria tend to have a strong codon usage bias in ribosomal protein genes to enhance ribosome assembly[20], as such, contigs cannot be accurately binned. Thus, the absence of ribosomal protein genes from MAGs caused by assembly is a hallmark

of fast-growing bacteria[21]. In addition, benefiting from enhanced metabolic efficiency and replication, organisms with smaller genomes gain a competitive edge in resource-limited settings, which is suggested by streamlining theory[22], and our observation of the dominance of small-genome bacteria supports this theory. Taken together, these findings indicate that the presence of nematodes accelerated the depletion of organic matter, leading to a shift in the dominant bacterial species towards those with smaller genomes at the late stage of carcass decomposition.

Despite significant changes in microbiota composition from the early to the late stage, certain bacteria persisted across both phases, with some species undergoing different succession patterns at the strain level (Supplementary Fig. 3c). Comparative genomic analyses of these strains uncovered potentially linked genes to these shifts. For instance, a *Serratia marcescens* strain with more genes related to virulence and competition, was less affected by *P. pacificus* at the late stage compared to another *S. marcescens* strain. Interestingly, when

*P. pacificus* was present, the abundance of *Empedobacter brevis* and various *Microbacterium* species increased at both stages compared to the nematode-free condition. Only a few known virulence genes have been identified in these MAGs, suggesting that they are potential candidates for investigation into pathogenicity of these bacteria to nematodes.

## Nematode physiology responds to natural microbiota

To understand how natural microbiota affects the nematodes' physiology at the early and late stage of succession, we conducted a Gene Set Enrichment Analysis (GSEA) using RNA-seq data from mixed-stage *P. pacificus* grown on the standard laboratory food source *Escherichia coli* (*E. coli*) OP50 and decomposed rose chafer grub carcasses. Our results showed that wild-type animals grown on grub carcasses had multiple amino acid synthesis and translation pathways downregulated compared to animals grown on agar plates with OP50 bacteria (Fig. 3a). Moreover, pathways related to propanoate metabolism and fatty acid degradation were also downregulated. In contrast, genes involved in defence response to bacteria were upregulated (Supplementary Data 2) on grub carcasses, indicating an antagonistic interaction with the bacterial community. At the late stage of the decomposition, when the nutrient quality of bacteria on the grub carcasses declined, pathways associated with genome replication, transcription, and protein synthesis/degradation were found to be further downregulated compared to standard OP50 conditions (Fig. 3a). In comparison to wild type *P. pacificus* animals, the cellulase-null mutants showed an even stronger downregulation of metabolic pathway genes (Fig. 3a). Notably, at the early stage, in the cellulase-null mutant energy synthesis pathways were downregulated, including the citrate cycle and oxidative phosphorylation, which were only observed in the wild type at the late stage. These findings might indicate that the cellulase-null mutant has a reduced ability to harvest nutrients from the microbiota. Additionally, when comparing *P. pacificus* across different decomposition stages, it was shown that multiple pathways involving amino acid metabolism and developmental signalling were downregulated at the late stage. In the cellulase-null mutant, the metabolism of xenobiotics by cytochrome P450 was upregulated, which might be a result of the accumulation of toxin from microbiota (Supplementary Fig. 4a, b). Finally, comparisons between the wild type and the cellulase-null mutant at both early and late stages revealed that the cellulase-null mutant has pathways including oxidative phosphorylation that were downregulated (Supplementary Fig. 4c, d).

We further investigated the gene expression changes in a stage-specific population, comparing young adults of *P. pacificus* wild type and cellulase-null mutant grown on grub carcasses of the white-spotted flower chafer at the early stage with those grown on *E. coli* OP50. We found that most enriched KEGG pathways in mixed-stage populations also overlapped with those found in stage-specific populations: among the 9 downregulated pathways in stage-specific population, 7 were also downregulated in mixed-stage wild type; similarly, 14 out 18 downregulated pathways were found in cellulase-null mutant population, and stage-specific populations showed more KEGG pathways that were downregulated (Supplementary Fig. 5a). Cross comparisons between the two nematode strains grown on grub carcasses showed several metabolic and developmental pathways downregulated in the cellulase-null mutant (Supplementary Fig. 5b). Additionally, our DESeq2 analysis revealed a higher number of differentially expressed genes in the stage-specific populations. This is likely due to the variability observed in mixed-stage populations (Supplementary Fig. 6). Finally, wild-type nematodes from grub carcasses at the early stage had a larger brood size than the cellulase-null mutant (median 214 vs. 157, $n = 15$; $p$ value = 0.012; Supplementary Fig. 5c). Therefore, the observed changes in the physiology of nematodes were likely caused by similar microbiota derived from the two chafer grubs rather than the age heterogeneity in tested nematode populations.

Nonetheless, it is important to note that the grub carcasses create an environment which had similar effects on the wild type and mutant nematodes, resulting in massive gene expression changes when compared to standard laboratory cultures grown on OP50 (Fig. 4). Specifically, at the early stage, the wild type and cellulase-null mutant displayed differential expression in 3643 and 5886 genes, respectively. At the late stage, this number increased to 10,787 and 11,873 genes for the wild type and cellulase-null mutant, respectively.

## Natural microbiota change fat metabolism in *P. pacificus*

Lipid droplets are often used as a general indicator of the metabolic state of nematodes[23]. We found that natural microbiota could greatly affect lipid distribution. Under standard laboratory conditions, *P. pacificus* adults feeding on *E. coli* OP50 accumulated large amounts of lipid droplets in intestinal and hypodermal cells, while on grub carcasses *P. pacificus* generally had a reduced level of lipid droplets (Fig. 3b). This observation is consistent with our transcriptomic analyses indicating that lipid metabolism genes were differentially expressed on grub carcasses. At the early stage of the decomposition, *P. pacificus* wild-type adults stored lipid droplets mainly in oocytes and eggs, while lipid droplets in the cellulase-null mutant were also found in the intestine (Fig. 3b). Moreover, the cellulase-null mutant showed more fat storage at the early stage of the composition, suggesting that these nematodes have a reduced level of interaction with the microbiota due to reduced foraging efficiency. In contrast, at the late stage of the composition, both wild type and cellulase-null mutant animals stored lipid droplets in the intestine with less extensive staining in the tail region compared to worms grown on OP50. Along with our transcriptomic analysis, these results indicate that the natural microbiota can affect the physiology of *P. pacificus* through lipid metabolism and alter the strategy of energy utilisation.

## Absence of *P. pacificus* cellulases induces dauer formation

The dauer stage is crucial for nematode ecology, characterised by its long lifespan, stress resistance, and ability to disperse over long distance[24]. At the late stage, we found an increase in the number of dauer larvae on late grub carcasses, likely associated with the decrease in the quality and quantity of bacterial food. We observed a higher proportion of dauer larvae in the cellulase-null mutant compared to wild-type animals ($p$ = 0.0012, Fig. 3c), and this increase in dauers is a sign of nematodes transferring from the reproductive to survival phase. Accordingly, the genes of target of rapamycin (TOR or mTOR) pathway were downregulated in the cellulase mutant (Fig. 3a), which is known to regulate the dauer formation[25]. The cellulolytic ability of *P. pacificus* has been shown to allow the digestion of cellulolytic biofilms and convert cellulose into simple sugars[14]. Interestingly, we observed an increase in the abundance of several putative pathogenic species, including *Microbacterium* sp. and *Leucobacter* sp., in the cellulase-null mutant treatment (Supplementary Fig. 3), suggesting that cellulases can help the nematodes suppress potential pathogens through disrupting biofilm formation. Therefore, food limitation and the increase in potential pathogens could be the primary factors inducing the formation of dauer larvae in these strains. These findings build upon our understanding from previous studies, suggesting that the cellulolytic ability could help *P. pacificus* acquire nutrients from natural microbiota and play a critical role in its survival in insect-decaying environments[14].

## Species-specific genes are more responsive to natural microbiota

All organisms possess species-specific genes that are thought to have evolved recently, maybe in response to the ecological niche of the organism[26]. *P. pacificus* has ~28,000 protein-coding genes, but only roughly one-third share a one-to-one orthology relationship with genes of *C. elegans*[27]. Approximately 15,000 protein-coding genes in *P.*

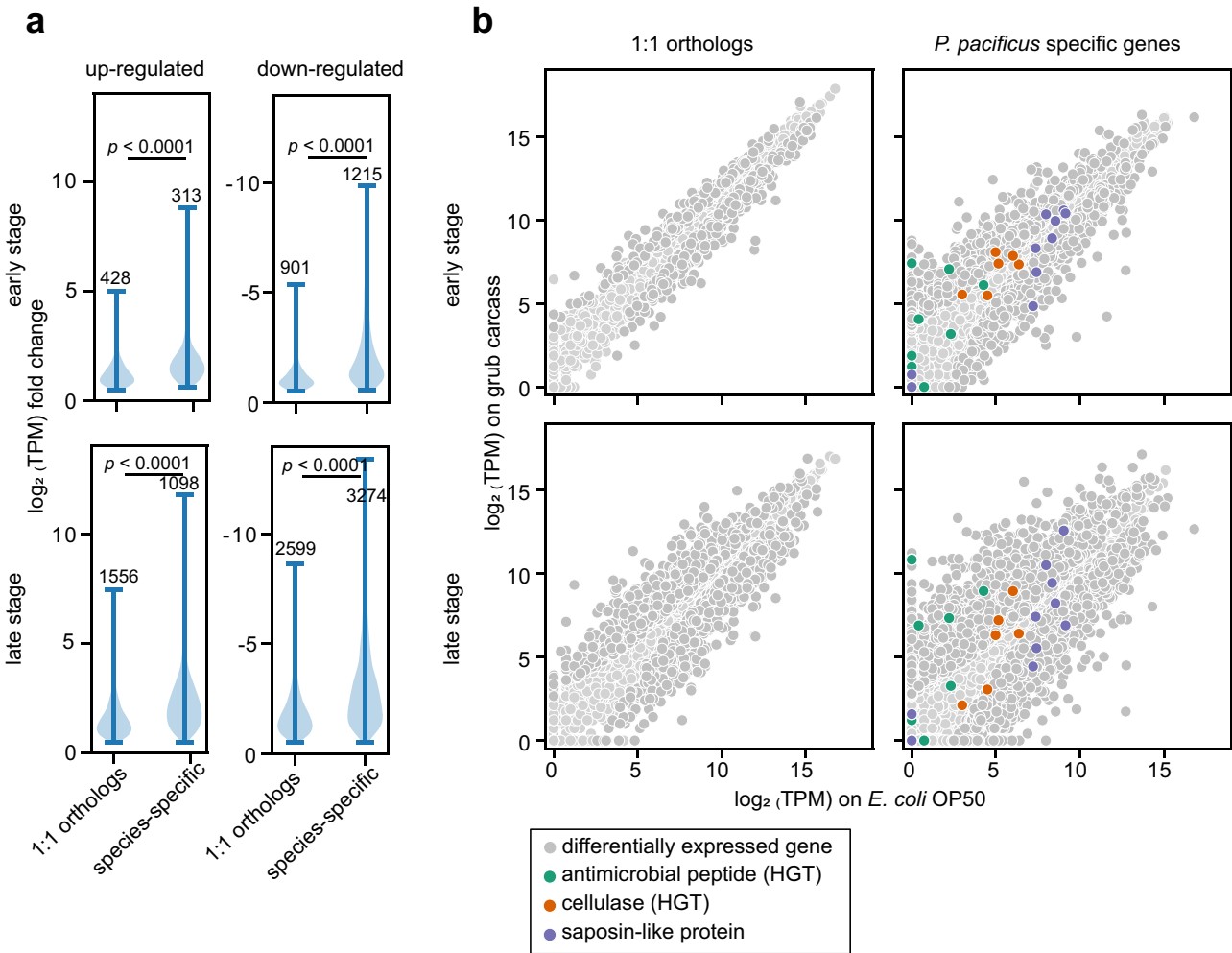

**Fig. 4 | Transcriptional profiling of *Pristionchus pacificus* wild type on different food resources at the early and late decomposition stage. a** Violin plot on the right illustrates the distribution of fold change for up- and downregulated genes in each category across two stages. The numbers indicate the gene count in each category. Statistical analyses were performed using a one-tailed *t*-test, with *p* < 0.0001 for all comparisons. **b** Gene expressions of *P. pacificus* 1:1 orthologs to *C. elegans* and *P. pacificus* species-specific genes on grub carcasses vs. *E. coli* OP50. Saposin-like genes, horizontally acquired cellulases, and diapausin-related antimicrobial peptides are highlighted.

*pacificus* are not found in *C. elegans*, making these species-specific genes particularly intriguing for evolutionary studies[12,13,28,29]. Importantly, however, many *P. pacificus*-specific genes are not associated with any phenotype under standard laboratory conditions[26], raising the question whether these species-specific genes have true biological significance or if they are merely rapidly evolving DNA sequences.

Our results revealed that species-specific genes in *P. pacificus* exhibit significantly larger expression fold changes compared to those genes with one-to-one orthologs in *C. elegans* on grub carcasses compared to *E. coli* OP50 controls (Fig. 4a, *p* < 0.0001). Within both species-specific genes and one-to-one orthologs, the number of downregulated genes outnumbered those that were upregulated. In *P. pacificus*, some species-specific gene families have gone through expansion during evolution, including saposin-like proteins, horizontally acquired cellulases, and diapausin (Fig. 4b). These genes are likely involved in interacting with bacteria. The differences in fold change of individual gene copies suggest functional diversification after gene duplication. For example, of the eight annotated diapausin genes, five had expression values close to 0 under standard laboratory conditions, whereas their expression increased from ~4- to 8200-fold when grown on grub carcasses. These results indicate that these genes are under different regulatory networks after gene expansion. Together,

these data suggest that species-specific genes do respond to the natural environment and might be of evolutionary importance in adaptation.

## Discussion

Soil nematodes are faced with a range of environmental challenges. Microbiota play different roles when interacting with nematodes including food, mutualistic bacteria, commensal, and pathogens. Additionally, the dynamics of the microbiota structure add another layer of complexity to understand these interactions. In particular, the succession of microbes in decaying organic matter is difficult to study in the un-manipulated environment due to uncontrolled starting material and variation of infestation rates. Our experimental setting using semi-artificial microcosm ecosystems allowed insights into the role of natural microbiota in nematode physiology. Thus, our study aimed to address a previous knowledge gap by using decomposed beetle grubs to simulate the microbial ecosystem and applying metagenomics sequencing to capture the functional dynamics of microbiota at different stages of decay. Moreover, we evaluated the corresponding physiological responses of wild type and mutant *P. pacificus* nematodes during these stages. Our study is the first characterisation of *Pristionchus*-associated microbiota, their succession, and impact on nematode physiology.

Our study results in three major findings. First, we provide insights into the food sources of *P. pacificus* in the natural environment and the nematode's influence on bacterial succession. At the early stage of the decomposition, nematodes were attracted to bacteria abundant in cofactor synthesis genes, e.g. B vitamins. Vitamin B12 has been shown to affect the life history traits of *C. elegans*[30,31], and also enhance the killing behaviour of *P. pacificus*[32], underscoring the potential fitness boost from acquiring vitamins. As decomposition progresses, bacteria with complex synthesising abilities were outcompeted by those with smaller genomes due to nematode predation and the decline of environmental nutrients. This observation fits the streamlining theory, according to which bacteria with smaller genomes are able to replicate and divide more rapidly giving them a competitive advantage in resource-limited environments[33]. Our results highlight the drastic changes in the dominant species of microbiota and their metabolic capacities during the succession (Fig. 1b, c).

Second, our study demonstrates that the succession of microbiota influences two key nematode metabolic strategies, fat utilisation and dauer formation, which are integral to survival and reproductive success. Previous studies showed that specific bacteria, including those producing vitamin B12 and *Lysinibacillus* sp.[3,32], can alter energy allocation from fat storage to reproduction. Consistent with these findings, we observed the presence of these bacteria in the early stage of nematode–microbiota interaction, highlighting their potential role in shaping nematode metabolism. In contrast, during the late stage of decomposition, increased fat storage may occur in response to reduced food quality and stress[23]. We also shed light on the regulation of dauer formation in *P. pacificus*, a mechanism influenced by integrated signals such as population density, food supply, and temperature in *C. elegans*[34]. Unlike *C. elegans*, *P. pacificus* shows a broader temperature tolerance[35] and varying dauer-inducing capacities among strains via pheromones[36]. Notably, the TGF-β signalling pathway, pivotal in *C. elegans*' dauer formation, may not hold the same importance for *P. pacificus*[37]. Instead, cues for *P. pacificus* to enter the dauer stage could stem from deficiencies in the vitamin B family and pathogen presence, possibly through the mTOR pathway signalling.

Lastly, species-specific genes are considered to be an evolutionary consequence of an organism's adaptation to its local environment, and they are of great interest for evolution. In *P. pacificus* most of the species-specific genes are not associated with any phenotype under the laboratory condition using the standard *E. coli* OP50 as food, but the expression levels of these genes are different on different monoxenic bacterial cultures[38]. Here, our data show that the expression of many species-specific genes changed substantially in response to carcasses decomposition, suggesting a potential role of these genes in more natural environments of *P. pacificus*. Though some may not hold biological significance, our data provide a resource for future functional analyses of species-specific genes. One kind of species-specific genes are those acquired through horizontal gene transfer. In our experiments, although the horizontally acquired cellulolytic ability had a limited effect on the overall structure of microbiota, it affected the abundance of several bacterial strains, resulting in an increased fitness in nematodes. These results support a role of horizontal gene transfer as a driving force in the evolution of eukaryotes. Along with other species-specific genes, they might not only assist in degrading cellulosic biofilms for additional carbon sources[14], but could also help nematodes tap into otherwise inaccessible valuable food sources within a natural microbiota, and suppress potential pathogens.

Decomposed grubs provide more than just a diverse array of microbes. They might contain natural stimuli that are crucial for shaping biological responses in nematodes. We speculate that the similar KEGG pathway enrichments observed in *P. pacificus* raised on rose chafer and white-spotted flower chafer grubs can be attributed to comparable microbiota profiles. However, the nutrient composition of decomposed grubs, which is different from that of standard NGM plates, is likely a contributing factor to the physiological alterations observed in nematodes. The bacterial species and their genetic diversity, together with bacteria's physiological responses to varying nutritional conditions, could both affect nematodes. Furthermore, nematodes grown in decomposed grub environments could achieve population densities beyond what is typically seen in lab settings. This phenomenon, possibly influenced by the intrinsic microbiota, may have implications for developmental processes and nematode behaviour. Therefore, the physiological modifications and regulatory changes observed in nematodes are probably linked to these comprehensive environmental influences, highlighting the complex interplay between microbiota composition, nutrient availability, and nematode physiology.

Prior studies in *C. elegans* have attempted to explore interactions between naturally isolated bacteria and the host[39–41]. However, *P. pacificus* represents a distinct ecology, and here our metagenomic-based experimental results serve as a genomic reference for isolating bacteria of interest in the future and expands our knowledge of the nematode–microbe relationship. Most importantly, this study emphasises the significance of studying host physiology within the context of its natural microbiota for advancing our comprehension of its biological complexity and potential for evolution. Thus, our study highlights the significant microbial succession on decaying organic matter and its reciprocal interaction with nematodes.

## Methods

### *P. pacificus* growth on bacterial community derived from grub carcases

Rose chafer (*C. aurata*) grubs were purchased from a pet supply store (Proinsects, GmbH, Germany), and white-spotted flower chafer (*P. brevitarsis*) grubs were acquired from a commercial insect farm in Anhui, China, to have, as much as possible, a consistent microcosm with limited beetle natural variation and small abiotic effects on beetle growth. To prepare grub carcasses, healthy grubs with an average weight of 2 g were soaked into water containing 0.01% Triton X-100 for 1 min to remove mites and nematodes, if any. They were rinsed with fresh water until the residual detergent was eliminated. These grubs were then dried with clean paper towels and sectioned using a sterile surgical razor through the longitudinal body plane. Sections from one given grub were placed onto a 1.7% water agar supplemented with the KPO buffer[42]. Those agar plates were sealed with parafilm and incubated in dark at 20 °C for 3 days before use.

*P. pacificus* wild type (laboratory standard strain PS312) and the cellulase-null mutant (Sommer lab strain RS3762[14]) strains were reared on the standard NGM plates with *E. coli* OP50. All *P. pacificus* strains were acquired from Ralf Sommer Lab at the Max Planck Institute for Biology, Germany. Non-starved *P. pacificus* nematodes were washed off the plate using the M9 buffer containing 0.01% Triton X-100[42]. Three more washes were performed to minimise the introduction of *E. coli* OP50 into the grub carcass microbiota. Around 500 individuals of mixed-staged wild type and cellulase-null mutant were used to inoculate the 3-day old grub carcass, respectively. As control, we used plates with grub carcasses without adding any nematode. We collected three samples from each treatment.

To collect stage-specific *P. pacificus* individuals from decomposing grub carcasses, ~800 synchronised second stage of juveniles (J2s) of wild type and cellulase mutant were prepared to inoculate 3-day old white-spotted flower chafer (*P. brevitarsis*) grub carcass plates following the above protocol. Seven days after inoculation, nematodes were extracted from plates and passed through a 100 and 40 μm nylon screen, where most adults were kept on the 40 μm screen. All nematodes were washed three times with M9 buffer. Small amounts of nematodes were used for 16S rDNA sequencing for nematode intestinal microbiota, while the rest of nematodes were prepared for RNA-seq experiments. In addition, young adult populations of wild type and

cellulase-null mutant were collected from 60 h after synchronised J2s under standard laboratory conditions using OP50 as the sole food source.

## DNA and RNA extraction

We constantly observed an increase in nematode population by day 7, marking a phase of visible abundance and nutrient richness. This stage was hence designated as the 'early stage' for sample collection. Subsequently, a significant shift towards the dauer stage, a phase of arrested development in nematodes, was consistently observed around the 4th week of decomposition. To capture the microbial and gene expression dynamics preceding this transition, samples were also collected on day 21, before the majority of nematodes enter the dauer stage. This time point is referred to as the 'late stage' in our study (Fig. 1a). Grub carcass plates were washed with M9 buffer with 0.01% Triton X-100 at the given sampling days. Solutions from the plate were collected into a 15 ml falcon tube and centrifuged at $200 \times g$ for 30 s to separate *P. pacificus* from bacteria. The supernatant containing bacteria was transferred to a new tube and concentrated at $2600 \times g$ for 30 min, and stored at −80 °C. Nematodes were washed twice with M9 buffer with 0.01% Triton X-100 to remove remaining bacteria and also stored at −80 °C.

A Zymo Direct-zol RNA Miniprep (Zymoresearch, Cat. No. R2053) kit was used to extract RNA from *P. pacificus* following the manufacturer's protocol. Bacterial cells and adult nematodes (for 16S rDNA sequencing) went through three freeze-thaw cycles using liquid nitrogen, and DNA was extracted using MasterPure™ Complete DNA and RNA Purification Kit (Lucigen, Cat. No. MC85200) and TaKaRa MiniBEST Universal Bacteria Genomic DNA Extraction Kit Ver. 3.0 (Takara, Cat. No. 9763) following the manufacturers' instructions. The bacterial metagenomics library and RNA-seq library preparation and sequencing were done by the company Novogene.

## Assembly and functional annotation of shotgun metagenomic sequences from the grub carcass

We first combined all raw reads from the same time point (nine each) and assembled them into contigs using metaSPAdes version 3.15.3[43] with default parameters. We kept contigs that were longer than 3 kbps. The raw reads from each metagenomics library were then separately mapped to the assemblies using HISAT2 v2.1.0[44]. MetaBAT2 v2.12.1[45] was used for the calculation of coverage and contigs binning. The average of the raw reads coverage of each bin was used to calculate the relative abundance of each MAG. For functional annotation, we predicted protein-coding regions of each bin using MetaProdigal[46], and acquired functional annotation by blasting against the non-redundant data set of bacterial sequences using DIAMOND[47]. The KEGG classification was acquired using eggNOG-mapper V2[48]. Finally, we applied BUSCO v5.3.2[49] to measure the completeness of the single-copy marker genes as the proxy of the integrity of assembled genomes. Furthermore, these single-copy marker genes were subjected to BLAST analysis, the results of which were utilised to assign species names to our MAGs. To find identical bacterial strains across different stages, MAGs bearing the same species name underwent pairwise BLAST analysis. MAGs were considered identical if nucleotide (nt) similarity exceeded 98% (Supplementary Data 1).

## 16S rDNA-based microbial communities quantification

To detect the nematode-associated microbiota, 16S rDNA (V4 region) sequencing was performed using the universal primers 515f (5′-TCGTCGGCAGCGTCAGATGTGTATAAGAGACAGGTGYCAGCMGCCG CGGTAA-3′) and 806r (5′- GTCTCGTGGGCTCGGAGATGTGTATAAG AGACAGGGACTACNVGGGTWTCTAAT-3′). Briefly, we used the software Mothur v1.48.0[50] following the MiSeq SOP[51] to cluster sequences into operational taxonomic units (OTUs) and assign taxonomy. Taxonomies were assigned to genus level using release 123 of the SILVA 16S

ribosomal RNA database. In addition, we reanalysed the 16S rDNA gene sequencing results from our previous microbiota studies on La Réunion Island[9]. The raw reads were acquired from NCBI BioProject accession number PRJNA698805 and analysed following the same procedure. We excluded rare OTUs present in less than four samples to eliminate potentially spurious identifications. Samples with fewer than 4000 reads were omitted from our analysis.

## Identification of *P. pacificus* antimicrobial peptides and transcriptome analysis

We used HISAT2 v2.1.0[44] to map raw reads to the *P. pacificus* reference genome (pristionchus.org, version: El Paco)[52]. We performed a reference-based transcriptome assembly using StringTie2[53], and the longest isoforms were kept for transcripts quantification using featureCounts[54]. The functional annotations of *P. pacificus* were assigned based on orthology with *C. elegans* using OrthoFinder[55]. For *P. pacificus* species-specific genes, the functions were predicted based on Pfam domain[56]. We used DeSeq2[57] to identify differentially expressed genes, and applied the GSEA[58] approach to detect the biological pathways that are differentially expressed via the KEGG gene set. A false discovery rate (FDR $q$ value) <0.05 was used. The outcomes of GSEA and the transcripts per million values derived from RNA-seq are comprehensively presented in Supplementary Data 2 and 3, respectively. The collection of gene sets was obtained from WormEnrichr[59], and the genes of *P. pacificus* were assigned into corresponding categories based on orthology with *C. elegans*. We used GSEAPy[60] to perform the analysis.

## Lipid droplet staining

We applied a lipid droplet staining protocol from *C. elegans* with optimisation[61]. *P. pacificus* was fixed in 1% paraformaldehyde/phosphate buffered saline (PBS) for 30 min, then immediately frozen in liquid nitrogen and thawed with running tap water. After three washing steps with 1× PBS, samples were dehydrated in 60% isopropanol for 2 min, then stained with 60% Oil-Red-O (Merck, Cat No. O0625) working solution for 30 min with rocking. Stained samples were washed three times with 1× PBS and mounted in 1× M9 and placed onto agarose-padded slides for imaging. The Oil-Red-O images were acquired on a Zeiss AxioImager Z1 microscope with an Axiocam 506 mono camera and edited using the software Fiji[62].

## Brood size assays

Non-gravid young adults reared on grub carcasses (*P. brevitarsis*) were singled out onto NGM plates with *E. coli* OP50 7 days after inoculation. Individual nematodes were transferred onto a new plate every day for 3 days and progeny were counted. Statistical analyses were performed using *t*-test.

## Dauer assays

For dauer assays, nematodes were maintained on decomposed grubs at 20 °C for 28 days. Subsequently, the nematodes were washed off the plates using distilled water, and a subsample was counted under a dissecting scope. An SDS assay was conducted onto a subsample to eliminate non-dauers for counting[63]. Each treatment was performed with three independent replicates.

## Reporting summary

Further information on research design is available in the Nature Portfolio Reporting Summary linked to this article.

# Data availability

Data generated in this article, including the raw reads of the metagenome of decomposed grubs, and raw reads of 16S rDNA from both decomposed grubs and the gut of nematodes, have been deposited in the NCBI database under the accession number PRJNA938905. The

RNA sequencing data can be found under accession number PRJNA938361. The gbff files for the MAGs have been deposited in Zenodo (https://doi.org/10.5281/zenodo.11196560). The 16S rDNA data generated by previous studies are sourced from NCBI database accession number PRJNA698805. Source data are provided with this paper.

## Code availability

Scripts and datasheets used to generate figures in this study are available in Zenodo (https://doi.org/10.5281/zenodo.10875885).

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

## Acknowledgements

The authors would like to thank Hanh Witte, Veysi Piskobulu, and Christian Weiler for technical assistance and Nick Youngblut for critical comments. This work was supported by grants from the National Natural Science Foundation of China (Grant No. 32200495, Z.H.; Grant No. 32370458, W.-S.L.), a Humboldt Research Fellowship for postdoctoral researchers from the Alexander von Humboldt Foundation (Z.H.), and the Max Planck Society.

## Author contributions

W.-S. L. and Z.H. designed and conducted the experiments, and performed analyses. R.J.S. supervised the project. W.-S. L., R.J.S., and Z.H. wrote the original draft. All authors have revised the manuscript and approved the submitted version.

## Competing interests

The authors declare no competing interests.
