## [Peer Review File · Nature Communications]

Microbiota Succession Influences Nematode Physiology in a Beetle Microcosm EcosystemREVIEWER COMMENTS

Reviewer #1 (Remarks to the Author):

The authors investigated the functional roles of the microbiome in the nematode model *P. pacificus* and its effects on host physiology, specifically in relation to lipid accumulation, dauer formation, and gene expression changes. This study was conducted within a controlled laboratory microcosm environment designed to emulate the process of insect carcass decomposition.

Notable insights from the study include:

1. Advance from previous research that primarily focused on changes in microbiome composition during the decay process, this study uses a metagenomic approach to explore the alterations in microbiome function. The results indicate an enrichment of B12 vitamin-producing microbes in high-nutrient conditions, while small-genome microbes dominate in low-nutrient environments.
2. The study provides evidence for the significance of cellulase in the nematode's survival in nutrient-limited and insect-decaying settings.
3. The research reveals that *P. pacificus*-specific genes exhibit significantly greater expression fold changes compared to shared orthologs with other nematodes, underscoring the nematode's capacity to respond and adapt to the microbiome.

While the manuscript presents new findings in the field of nematode microbiome research, there are concerns regarding missing background information, the rationale behind the study's design, detailed descriptions of the methods used, and potential issues in data analysis and interpretation.

Major concerns include:

1. The use of mixed-stage worms (Line 101), potentially causing gene expression variations due to different developmental stages. To determine whether these developmental stage variations need to be taken into account when interpreting the microbiome and transcriptome findings, it could be crucial to incorporate microscope images and quantitative data to clearly depict the developmental stage of the worms at the time of sampling.
2. The rationale for selecting day 7 and day 21 as time points for analysis is not explained. A more comprehensive, long-term study of microbiome composition may be necessary to justify these specific time points.
3. The source of the microbiome is not clearly defined, raising questions about whether it originates from beetles or introduced by the nematodes.
4. When comparing the gene expression of nematodes raised on carcasses to those cultivated on OP50, numerous variables are introduced, such as exposure to a wide array of microbes and the levels of available nutrient sources. This can complicate the interpretation of differentially expressed genes. It may be helpful to include two additional comparisons: 1. wild type versus mutant, and 2. the same genotype at an early stage versus a late stage, for a more comprehensive assessment of the dataset.
5. The selection of enriched pathways in differential gene expression analysis appears arbitrary and should be based on statistical criteria.

Minor concerns include:

1. The metagenome data only provides relative abundance, and it would be valuable to quantify the total microbiome abundance to better understand the impact of microbes and changes in diversity.
2. The observation of higher lipid accumulation in the intestines of nematode cellulose mutants and wild-type nematodes grown on OP50 is intriguing. Higher lipid accumulation in the intestine is a potential sign of reduced fecundity. To further explore this, quantifying progeny production for wild-type nematodes could provide valuable insights.
3. The study does not address whether there is microbiome colonization in the gut of nematodes and, if so, what these specific microbes are.
4. The study should explore the impact of cellulose gene deletion on nematode growth, as the differences in microbiome composition between the wild type and cellulose gene deletion mutant

may be influenced by the mutant's slower growth rate.

In conclusion, while the research exhibits significant contributions to the field of nematode microbiome, addressing these concerns will enhance the overall quality and reproducibility of the study.

Reviewer #2 (Remarks to the Author):

The manuscript from Lo, Sommer, and Han titled: "Microbiota Succession Influences Nematode Physiology in a Beetle Microcosm Ecosystem" is a nicely written study looking at microbial community succession in a host-microbe system within an insect decaying environment. The authors use metagenomic and metatranscriptomic to examine microbial community composition differences in decaying beetle grubs and the host gene transcription response. This study proposes a promising synthetic framework to perform controlled experiments to understand host-microbe interactions in a non-model organism. I am pleasantly surprised at how stable the microbial community associated with beetles bought from a random third-party vendor, opening many future possibilities with this system. Lastly, the observed reduction in average genome size over time, correlating with streamlining theory, is exciting.

Regarding the methodological approach. Overall, the methods are correct. However, it does not exploit the full potential offered by this type of data.

First, better methods than metagenomic reconstruction exist to profile microbial communities at the taxonomy and functional levels. If metagenomic reconstruction algorithms improved over the past years, they can only capture a limited fraction of the organisms in complex communities. Such methods are typically only capable of assembling metagenomes of the highly abundant or highly variable bacteria present in the community. Over half of the assembled contigs in the two combined metagenomic reconstructions have no bin assignment. I suggest using a Reference-based computational approach such as Metaphlan 4 (which includes an extended taxonomy coverage using the GTDB database), and Humaan 3 would provide the author with more detailed taxonomic and functional profiles. Metaphlan 4 can provide taxonomic profiles to the species level when possible, overcoming the limitation of 16S rRNA mentioned by the authors in the introduction.

Despite the previous comment, the authors generated 305 MAGs out of 18 samples, which is a very good yield. This precious genome-scale data could be used further:

- 1) From the methods, it appears that raw reads for one sample have been mapped to their associate time point. Running a mapping process with all the MAGs combined would be interesting. This remapping would allow the authors to track the presence of MAG predicted at one time point on the other.
- 2) Similarly, the genomic composition of MAGs assigned to the same genus/species could be compared across time points to see if the same bin has been predicted twice or if any specific functional change occurred. Several genera have some interesting transitions, especially Myroides (Are the late-stage Myroides present in the control samples the same as the one in the early stage?), Serratia, Brucella (is Brucella(93) the same as Brucella(249) and if not what are the differences), etc... Genome comparison could be done first by looking at Average Nucleotide Identity (ANI), then at gene/function comparison.
- 3) The authors mention that 16S rRNA does not allow species identification, yet no specific bacterial species besides several "sp." species are mentioned across the manuscript. The supplementary table does not even mention any species-level taxonomy. Taxonomic assignments could be done more accurately than blasting against Blast NR. Dedicated tools such as gtdb-TK do a fantastic job assigning bin/genome to specific species/genera.
- 4) Genomic Bin can be annotated better. The authors performed KEGG annotation using EggNog, but they could have a better overview of functional potential by using fast and generalist tools such as Prokka or Bakta. Such tools could provide annotations not covered by Kegg, especially when looking at virulence genes (such as toxin or invasion-related genes.)

Regarding the Transcriptomic approach:

- 1) Looking at Figures 3 and 4. Some genes appear to be positively over-expressed in the *P. pacificus* nematodes growing on the grubs. If correct, what are the genes in that category? This observation correlates with the violin plots of Figure 4 that indicate an overall positive over-expression in *P. pacificus* on grubs.
- 2) Comparing transcription profiles to a control population on OP50 is an important comparison to grasp how the environment influences overall expression. However, it could be interesting to also display the differences between time points of the same environment, especially in the later days.
- 3) The authors should look for correlations between host transcriptome expression and microbial community composition. It could be interesting to see if any link between host and bacteria can be drawn from this experiment.

While I agree with the authors that the genomic composition of the bacteria in the environment will influence host physiology. I am not convinced the data presented in this study is enough to justify the author's statement that change in abundance is related to the host's preferred diet. The presence of the nematode could be a significant driver of environmental change. Worms induce mechanical shuffling of the microbial community and exude different waste products (diffusion of ammonia, defecation of organic matter) that could cause molecular changes in the environment and influence community composition. In addition, at the early stage, when the environment is rich in nutrients and the population of nematodes is low, bacterial growth may keep pace with nematode consumption. This balance is overturned when the nematode population reaches a critical mass and nutrients are depleted, drastically changing community composition. The authors noticed transcriptome expression changes in the cellulase-deficient worms they linked to inefficient foraging (line 243), a statement that the Dauer count and fat storage differences reinforce. However, the community analysis (PCoA plot Supp figure 1A) suggests no significant difference between worm-associated communities. Only a few bacteria rankings changed between the two strains, indicating that either 1) diet changes are not drastically influenced by cellulase or 2) Community composition is not predominantly affected by nematode diet.

In the second paragraph of the "*P. pacificus* species-specific genes are more responsive to the natural microbiota," the authors repeatedly mention gene duplications. Are they referring to the gene families mentioned in the paragraph (saposin-like proteins, horizontally acquired cellulases, and diapausin)? Or do they refer to all genes *P. pacificus* specific? In both cases, the section needs clarification.

Finally, more data should be shared with this publication. The authors generated such exciting data that it would be nice for other researchers to have access to all the analysis/work performed for this manuscript. I would encourage authors to submit the MAGs abundance/Coverage table and the transcript abundance table as supplementary tsv/excel files. As an NCBI/ENA submission or Figshare/Zenodo zip file(s), I would also submit the MAGs. The genomic information found associated with nematode could be helpful to other researchers. Lastly, the code used for generating MAGs and transcriptome analysis could also be a welcome addition.

Minor comments

- The method used to compare communities with *la reunion* beetle 16S rRNA needs to be included.
- Line 108; 217 MAGs mentioned for late stage, Supp table list 216
- Line 329: *Lysinibacillus* spp., while mentioned previously as *Lysinibacillus* sp.
- Line 329 to 331: "with early stage interactions with vitamin B12 and *Lysinibacillus* spp. altering energy allocation from fat storage to reproduction." This statement appears a little bit out of nowhere and is not mentioned enough before to claim "bacteria and food availability governs fat utilization".
- Methods: Line 412 contigs greater than 3kbp are kept, but line 422 MAGs longer than 1kbp are kept. I guess it is 1Mbps? In this case, it contradicts line 109, which states that 1.5 Mbps MAGs were kept.
- Legends: Bot instead of dot? (lines 619 and 666)
- Figure 1 B. Maybe adjusting the color in the stacked bar plot to taxonomy could help grasp significant differences between time points. (For example, the Gammaproteobacteria could be different shades of red).
- Figure 2. I really like this figure. It is a nice way to convey abundance and functionality. The ranking panel needs to be clarified. As I read from left to right, I assume 1st on the left, but it

could be different for people reading from right to left. Adding a 1st/10th/20th on the x-axis could help read the figure.

Reviewer #1

The authors investigated the functional roles of the microbiome in the nematode model *P. pacificus* and its effects on host physiology, specifically in relation to lipid accumulation, dauer formation, and gene expression changes. This study was conducted within a controlled laboratory microcosm environment designed to emulate the process of insect carcass decomposition.

Notable insights from the study include:

1. Advance from previous research that primarily focused on changes in microbiome composition during the decay process, this study uses a metagenomic approach to explore the alterations in microbiome function. The results indicate an enrichment of B12 vitamin-producing microbes in high-nutrient conditions, while small-genome microbes dominate in low-nutrient environments.
2. The study provides evidence for the significance of cellulase in the nematode's survival in nutrient-limited and insect-decaying settings.
3. The research reveals that *P. pacificus*-specific genes exhibit significantly greater expression fold changes compared to shared orthologs with other nematodes, underscoring the nematode's capacity to respond and adapt to the microbiome.

While the manuscript presents new findings in the field of nematode microbiome research, there are concerns regarding missing background information, the rationale behind the study's design, detailed descriptions of the methods used, and potential issues in data analysis and interpretation.

Major concerns include:

1. The use of mixed-stage worms (Line 101), potentially causing gene expression variations due to different developmental stages. To determine whether these developmental stage variations need to be taken into account when interpreting the microbiome and transcriptome findings, it could be crucial to incorporate microscope images and quantitative data to clearly depict the developmental stage of the worms at the time of sampling.

Response: We agree with the reviewer's concern that the developmental stage could be a confounding factor. To resolve this, we did additional RNA-seq experiments to compare transcriptomic profiles of *P. pacificus* young adults reared on grub carcasses to young adults grown on *E. coli* OP50 under standard laboratory conditions.

We conducted additional experiments using the white-spotted flower chafer (*Protaetia brevitarsis*) grub, which shares a detritivorous diet with the rose chafer grub in their larval stages and belong to the same family, Scarabaeidae, within the order Coleoptera (beetles). This is because two major authors W-S. L. and Z. H., have relocated from Max Planck Institute, Germany, to Northwest A&F University, China. The rose chafer grub (*Cetonia aurata*) is a regional pest in Europe and not commercially available in China.

To prepare stage-specific *P. pacificus* from grub carcasses, we inoculated the decomposed grub carcass (*Protaetia brevitarsis*) with synchronized second stage juveniles (J2s). After 7 days, the majority of nematodes are the second generation young adults. Nematodes were passed through nylon screens with pore sizes of 100 μm and 40 μm , effectively retaining most adults on the 40 μm screen. We showed that most changed KEGG pathways in mixed-stage populations were also found in young adult populations (Supplementary Fig. 5a,b). Additionally, the mixed-stage populations revealed fewer differentially expressed genes detected by DESeq2 (Supplementary Fig. 6), and fewer enriched pathways. This is because of the variability of expression profiling among the biological replicates, which may be caused by divergent age structure. Thus, the mechanisms conserved across different age groups were predominantly captured in mixed-stage populations. This indicates that the changes we observed are robust across developmental stages, highlighting conserved regulatory mechanisms rather than stage-specific responses. We excluded the comparison at the late stage of grub decomposition, due to the heterogeneity of the nematode population structure. We have added these results in the revised version.

2. The rationale for selecting day 7 and day 21 as time points for analysis is not explained. A more comprehensive, long-term study of microbiome composition may be necessary to justify these specific time points.

Response: Our pre-trial observations indicated that by day 7 their population increased to a level easily detectable through visual assessment. This stage was chosen as the early stage for sample collection. Moreover, our observations revealed the presence of a significant amount of dauer larvae by day 28. Dauer is a non-feeding, arrested stage with a presumably reduced impact on the microbiome. Therefore, we opted for day 21 to observe the onset of the dauer stage to explore the microbial composition and gene expression patterns associated with the transition of worms into the dauer stage. The rationale behind these specific time points is clarified in the “DNA and RNA extraction” section in Materials & Methods.

3. The source of the microbiome is not clearly defined, raising questions about whether it originates from beetles or introduced by the nematodes.

Response: We are sorry for the confusion here. Both *P. pacificus* wild type and cellulase-null mutant were reared on a monoxenic culture using *E. coli* OP50 as the sole food source before using them in the experiment. Thus, no bacteria other than *E. coli* could come from the nematodes. In the experimental setup, nematodes were washed with sterile M9 buffer before inoculation to minimise the amount of OP50 being introduced to the microbiota. Consequently, our analysis did not show the existence of OP50 at either sampling point. In addition, grubs were washed before experiments. Thus, the microbiome mainly originated from beetle grub carcasses, rather than being introduced by the nematodes. We have clarified this in Materials & Methods.

4. When comparing the gene expression of nematodes raised on carcasses to those cultivated on OP50, numerous variables are introduced, such as exposure to a wide array of microbes and the levels of available nutrient sources. This can complicate the interpretation of differentially expressed genes. It may be helpful to include two additional comparisons: 1. wild type versus mutant, and 2. the same genotype at an early stage versus a late stage, for a more comprehensive assessment of the dataset.

Response: We have added those comparisons (supplementary Fig. 4) and described those in the results. We found that 1) cellulase-null mutant had pathways including ribosome, oxidative phosphorylation and citrate cycle downregulated at the early stage, while ribosome and oxidative phosphorylation downregulated at the late stage; 2) either wild type or mutant had multiple pathways downregulated involving metabolism and development at the late stage. We have added these results in the text.

5. The selection of enriched pathways in differential gene expression analysis appears arbitrary and should be based on statistical criteria.

Response: These are in fact the statistically significant results (FDR q value < 0.05; Fig. 3a). We only found pathways that were downregulated on grub carcasses vs. under standard lab conditions (*E. coli* OP50). This trend can be attributed to a greater number of genes being downregulated in this environment. We updated the number of up- and down-regulated genes in Fig. 4. In additional comparisons suggested by reviewers, we found a few pathways that were upregulated (Supplementary Fig. 4,5).

Minor concerns include:

1. The metagenome data only provides relative abundance, and it would be valuable to quantify the total microbiome abundance to better understand the impact of microbes and changes in diversity.

Response: We understand the limitation of not having absolute abundance data in our study and appreciate the reviewer's attention to this aspect. To mitigate it, we adopted a

ranking-based approach in our analysis, focusing on the shifts in rankings of various MAGs under different treatment conditions. This method enabled us to conservatively interpret the changes in relative abundance without the need for absolute quantities.

Our study primarily aimed to decipher the bacterial genomic composition in various treatments and to understand how these genomic characteristics influence nematode physiology. Nonetheless, we agree that exploring absolute abundance, such as determining the minimum quantity of beneficial bacteria in a community required to effectively alter nematode physiology, could be a valuable avenue for future investigations. This could provide deeper insights into the intricate relationships between microbial communities and host organisms.

2. The observation of higher lipid accumulation in the intestines of nematode cellulose mutants and wild-type nematodes grown on OP50 is intriguing. Higher lipid accumulation in the intestine is a potential sign of reduced fecundity. To further explore this, quantifying progeny production for wild-type nematodes could provide valuable insights.

Response: We have conducted an additional experiment which singled out non-gravid adults of *P. pacificus* wild type and cellulase-null mutant from grub carcasses to standard OP50-NGM plates. We counted progeny for day 1, day 2 and, day 3 and after. Our results showed that the wild type nematodes produce more progeny than the cellulase-null mutant when reared on grub carcasses. (Supplementary Fig. 5c)

3. The study does not address whether there is microbiome colonization in the gut of nematodes and, if so, what these specific microbes are.

Response: A notable and important anatomical aspect of *Pristionchus* is the absence of a grinder, an organ many nematodes use to break down bacteria. Consequently, live bacteria can pass through the pharynx and are long known for accumulating in the gut without being mechanically disrupted. To address the reviewer's question, we carried out targeted experiments using synchronized J2 stages of both nematode strains, which were inoculated onto 3-day old grub (*Protaetia brevitarsis*) carcasses. At day 7 post-inoculation, we collected both the decomposing grub biomass and the nematodes. Nematode samples were washed three times with M9 buffer with 0.1% Triton X-100 to remove any microbes on their surface. The DNA was then extracted, and the 16S V4 region was sequenced to analyze the microbial compositions of the decomposing grub and the nematode gut.

Our 16S rDNA profiling revealed that the microbiota within *P. pacificus*' gut resembles those species from the grub carcass, which is consistent with *Pristionchus*' anatomy and digestion mechanism. We did observe differences in the relative abundance of

various bacterial species (Supplementary Fig. 2c,d). This could potentially be explained by the nematodes' selective feeding on specific bacterial species.

4. The study should explore the impact of cellulose gene deletion on nematode growth, as the differences in microbiome composition between the wild type and cellulose gene deletion mutant may be influenced by the mutant's slower growth rate.

Response: We think the reviewer was intended to ask if the microbiome could have an impact on the growth of *P. pacificus* cellulase-null mutant in comparison to the wild-type. However, an accurate assessment on the growth speed is difficult to perform on the grub carcass setting. Thus, we measured reproductive ability, another most important life history trait. Please see Minor concern 2 for details. We found that *P. pacificus* wild type has a significantly larger brood size than cellulase-null mutant (Supplementary Fig. 5c), and this finding is consistent with our previous results using *E. coli* K-12 producing cellulosic biofilms (Han et al., 2022).

In conclusion, while the research exhibits significant contributions to the field of nematode microbiome, addressing these concerns will enhance the overall quality and reproducibility of the study.

Reviewer #2

The manuscript from Lo, Sommer, and Han titled: "Microbiota Succession Influences Nematode Physiology in a Beetle Microcosm Ecosystem" is a nicely written study looking at microbial community succession in a host-microbe system within an insect decaying environment. The authors use metagenomic and metatranscriptomic to examine microbial community composition differences in decaying beetle grubs and the host gene transcription response. This study proposes a promising synthetic framework to perform controlled experiments to understand host-microbe interactions in a non-model organism. I am pleasantly surprised at how stable the microbial community associated with beetles bought from a random third-party vendor, opening many future possibilities with this system. Lastly, the observed reduction in average genome size over time, correlating with streamlining theory, is exciting.

Regarding the methodological approach. Overall, the methods are correct. However, it does not exploit the full potential offered by this type of data.

First, better methods than metagenomic reconstruction exist to profile microbial communities at the taxonomy and functional levels. If metagenomic reconstruction algorithms improved over the past years, they can only capture a limited fraction of the organisms in complex communities. Such methods are typically only capable of assembling metagenomes of the highly abundant or highly variable bacteria present in

the community. Over half of the assembled contigs in the two combined metagenomic reconstructions have no bin assignment. I suggest using a Reference-based computational approach such as Metaphlan 4 (which includes an extended taxonomy coverage using the GTDB database), and Humaan 3 would provide the author with more detailed taxonomic and functional profiles. Metaphlan 4 can provide taxonomic profiles to the species level when possible, overcoming the limitation of 16S rRNA mentioned by the authors in the introduction.

Response: We appreciate the reviewer's positive and valuable feedback. Initially, our analysis approach involved using reference-based software for microbial structure profiling. However, we encountered a notable issue of having a high proportion of unmapped raw reads. Specifically, using Kraken2, we observed approximately 20% of raw reads at the genus level and 40% at the species level remained unassigned. This trend persisted with the use of HUMAnN 3.0 and the database `full_chocophlan.v296_201901.tar.gz` (as of December 2021), where a significant percentage of raw reads also remained unassigned. This challenge led us to opt for *de novo* assembly. This time, we updated to HUMAnN 3.8, utilizing the most recent database. Nevertheless, we still encountered about 30% and 50% of raw reads from early and late stage, respectively, that could not be assigned (i.e. UNMAPPED and UNINTEGRATED). Several abundant genera, e.g. *Myroides*, *Lysinibacillus*, and *Pseudochrobactrum* were not detected by HUMAnN.

For species assignment, we utilized Metaphlan-4.0.6 along with the database `mpa_vOct22_CHOCOPhlanSGB_202212`. Our findings indicated a comparable number of species detected between Metaphlan and the *de novo* assembly approach. In the early stage, across nine libraries, we identified between 78 to 94 species from the libraries, totaling 177. In the late stage, the species count ranged from 114 to 153, with a total of 243 distinct species. However, we also noticed some species demonstrated substantial raw reads coverage and BUSCO support identified by the *de novo* assembly was not identified by Metaphlan. In the interest of maintaining consistency in our results, we ultimately decided to rely on the findings from our *de novo* assembly.

We appreciate the reviewer's observation regarding the non-informative nature of our description of MAGs results. During our binning process, we discarded 50% of the contigs, most of which had low coverage. These are likely derived from less abundant bacteria or even eukaryotic microorganisms. After the removal of unbinned contigs and bins without BUSCO support, we observed that 76% and 63% of raw reads from the early and late stages, respectively, were mapped to our final MAGs. We have revised the corresponding sections of our results and updated the statistical data in Supplementary Figure 1 and Supplementary Data 1.

We acknowledge that our methods primarily detect relatively high abundant bacterial species. Therefore, this study discusses these highly abundant bacterial species, and we have emphasized this point in our manuscript.

Despite the previous comment, the authors generated 305 MAGs out of 18 samples, which is a very good yield. This precious genome-scale data could be used further:
1) From the methods, it appears that raw reads for one sample have been mapped to their associate time point. Running a mapping process with all the MAGs combined would be interesting. This remapping would allow the authors to track the presence of MAG predicted at one time point on the other.

Response: We performed a cross-mapping of raw reads across different time points, and less than 3% of the reads could be mapped to MAGs predicted at alternate time points. Our comparative analysis of MAGs sharing the same species name but identified at different time points revealed differences in their orthologous amino acid sequences, indicating these MAGs are not identical strains. The small proportion of reads that could be cross-mapped likely represents conserved genomic regions that are common across different bacterial strains.

2) Similarly, the genomic composition of MAGs assigned to the same genus/species could be compared across time points to see if the same bin has been predicted twice or if any specific functional change occurred. Several genera have some interesting transitions, especially *Myroides* (Are the late-stage *Myroides* present in the control samples the same as the one in the early stage ?), *Serratia*, *Brucella* (is *Brucella*(93) the same as *Brucella*(249) and if not what are the differences), etc... Genome comparison could be done first by looking at Average Nucleotide Identity (ANI), then at gene/function comparison.

Response: We thank the reviewer for valuable suggestions regarding our taxonomic assignments, leading to improved species resolution in our study. In our refined analysis, we discerned that *Brucella tritici* (93) and *Brucella pituitosa* (249) represent distinct bacterial species. Using this new species assignment, we identified 33 bacterial species that persisted across both early and late time points. The shared orthologous protein-coding sequences amongst these species were not identical, suggesting the presence of different strains rather than identical isolates. This strain diversity within the same species is exemplified in the Supplementary Fig. 3c. It highlights 12 abundant species. Taking *Myroides odoratus* and *Serratia marcescens* for example, these species were predominant across all treatments at both stages but showed a reduction in abundance after the introduction of two nematode genotypes at the late stage. This observation suggests a possibility of multiple bacterial strains of the same species co-existing, potentially filling similar ecological niches.

3) The authors mention that 16S rRNA does not allow species identification, yet no specific bacterial species besides several "sp." species are mentioned across the manuscript. The supplementary table does not even mention any species-level taxonomy. Taxonomic assignments could be done more accurately than blasting against Blast NR. Dedicated tools such as gtdb-TK do a fantastic job assigning bin/genome to specific species/genera.

Response: In addition to utilizing eggNOG for KEGG categorization, we also ran DIAMOND blast against the NCBI Refseq database to acquire functional annotation. We applied the results for the identification of bacterial species. We have further refined our methodology to mitigate the inaccuracies in species assignment due to horizontal gene transfer. To achieve this, we used the housekeeping gene set identified by BUSCO as the basis for species assignment. Consequently, we excluded MAGs with low BUSCO support. The updated results have been incorporated into Supplementary Data 1, along with revised methods in our study.

4) Genomic Bin can be annotated better. The authors performed KEGG annotation using Egnog, but they could have a better overview of functional potential by using fast and generalist tools such as Prokka or Bakta. Such tools could provide annotations not covered by Kegg, especially when looking at virulence genes (such as toxin or invasion-related genes.)

Response: We acknowledge the importance of identifying virulence genes, particularly those associated with nematode pathogenicity. However, our understanding of the pathogen in *P. pacificus* natural habitats is limited, contributing to the scarcity of studies on pathogenicity to *P. pacificus*. For instance, *Bacillus thuringiensis*, a known nematode pathogen, appears to have no effect on *P. pacificus*. In addition, we did not identify known toxins such as *B. thuringiensis* bacteria crystal (Cry) and cytolytic (Cyt) toxins in our MAGs. Nevertheless, we have documented the presence of bacterial secretion systems types 3, 4, and 6, which are often implicated in the injection of toxins in animal and plant pathogens, or microbial competition.

Regarding the Transcriptomic approach:

1) Looking at Figures 3 and 4. Some genes appear to be positively over-expressed in the *P. pacificus* nematodes growing on the grubs. If correct, what are the genes in that category? This observation correlates with the violin plots of Figure 4 that indicate an overall positive over-expression in *P. pacificus* on grubs.

Response: We apologize for any confusion caused by the original presentation in Fig. 4, which depicted the absolute values of fold changes, combining both upregulated and downregulated genes. We have modified Fig. 4 to separate the genes that are upregulated (left) from those that are downregulated (right), and add gene counts for

each category. Our analysis, supported by our GSEA results, which relies on different statistical approaches, reveals that there are more genes significantly downregulated on grubs.

2) Comparing transcription profiles to a control population on OP50 is an important comparison to grasp how the environment influences overall expression. However, it could be interesting to also display the differences between time points of the same environment, especially in the later days.

Response: We agree with the reviewer and this comment is consistent with Reviewer 1. We have added additional analyses which compare transcription profiles of a given strain at the early and late stage. These results were described in the context and displaced in Supplementary Fig. 4,5. Please also see Major concern 1 from the reviewer 1.

3) The authors should look for correlations between host transcriptome expression and microbial community composition. It could be interesting to see if any link between host and bacteria can be drawn from this experiment.

Response: We investigated the correlation between the changes in the abundance of MAGs and gene expression in two genotypes of *P. pacificus*. Our analysis revealed significant correlations between the abundance of 18 and 19 MAGs from the early and late stages, respectively, and the expression levels of 4,284 and 6,597 *P. pacificus* genes, after adjusting for false discovery rate. However, among these *P. pacificus* genes, only 174 and 172 had GO assignments, and we did not detect any significantly enriched pathway. The majority of these genes are specific to *P. pacificus* and their functions may be intricately linked to specific bacterial functions. This insight may guide future research, focusing on the specific roles of these unique genes in response to its microbial environment. We appreciate the reviewer's suggestion to explore new dimensions in our research.

While I agree with the authors that the genomic composition of the bacteria in the environment will influence host physiology. I am not convinced the data presented in this study is enough to justify the author's statement that change in abundance is related to the host's preferred diet. The presence of the nematode could be a significant driver of environmental change. Worms induce mechanical shuffling of the microbial community and exude different waste products (diffusion of ammonia, defecation of organic matter) that could cause molecular changes in the environment and influence community composition. In addition, at the early stage, when the environment is rich in nutrients and the population of nematodes is low, bacterial growth may keep pace with nematode consumption. This balance is overturned when the nematode population

reaches a critical mass and nutrients are depleted, drastically changing community composition.

Response: We agree with the reviewer's concern. We changed the subtitle "*P. pacificus* preferentially consume bacteria of high nutritional values" to "Nematodes contribute to vitamin B-producing bacteria decline" (line 134). We have reworded the statement in the following context to tune it down.

The authors noticed transcriptome expression changes in the cellulase-deficient worms they linked to inefficient foraging (line 243), a statement that the Dauer count and fat storage differences reinforce. However, the community analysis (PCoA plot Supp figure 1A) suggests no significant difference between worm-associated communities. Only a few bacteria rankings changed between the two strains, indicating that either 1) diet changes are not drastically influenced by cellulase or 2) Community composition is not predominantly affected by nematode diet.

Response: We agree with the reviewer that the cellulolytic ability of *P. pacificus* did not have a noticeable impact on the overall structure of the bacterial community, which is consistent with reviewer point 1. We have added this statement in line 121. The "key bacteria" change is associated with significant physiological change in the nematodes.

In the second paragraph of the "*P. pacificus* species-specific genes are more responsive to the natural microbiota," the authors repeatedly mention gene duplications. Are they referring to the gene families mentioned in the paragraph (saposin-like proteins, horizontally acquired cellulases, and diapausin)? Or do they refer to all genes *P. pacificus* specific? In both cases, the section needs clarification.

Response: We intended to state that *P. pacificus* species-specific gene families have gone through expansion rather than the whole genome duplication. In line: 292-293, we have added a sentence to clarify this point.

Finally, more data should be shared with this publication. The authors generated such exciting data that it would be nice for other researchers to have access to all the analysis/work performed for this manuscript. I would encourage authors to submit the MAGs abundance/Coverage table and the transcript abundance table as supplementary tsv/excel files. As an NCBI/ENA submission or Figshare/Zenodo zip file(s), I would also submit the MAGs. The genomic information found associated with nematode could be helpful to other researchers. Lastly, the code used for generating MAGs and transcriptome analysis could also be a welcome addition.

Response: We have included the MAGs abundance/coverage table and the transcript abundance table as Supplementary Data 1 and 3, respectively. Additionally, we have submitted our MAGs to the NCBI database. In our study, we utilized well-maintained software and analysis pipelines, all of which are accompanied by comprehensive

documentation for ease of use. Our code primarily involved the adaptation for execution on our high-performance computing infrastructure and basic format conversion tasks. We include a statement in the Data Availability section of our manuscript indicating that our specific code implementations are available upon request.

Minor comments

- The method used to compare communities with la reunion beetle 16S rRNA needs to be included.

Response: We have now included the method for the 16S rRNA community comparison in our manuscript. We thank the reviewer for pointing out this omission.

- Line 108; 217 MAGs mentioned for late stage, Supp table list 216

Response: We thank the reviewer for pointing out this discrepancy between the text and the Supplementary Table. We have updated the number of MAGs to ensure consistency in our documentation.

- Line 329: *Lysinibacillus* spp., while mentioned previously as *Lysinibacillus* sp.

Response: We have corrected the nomenclature from "*Lysinibacillus* sp." to "*Lysinibacillus* spp."

- Line 329 to 331: "with early stage interactions with vitamin B12 and *Lysinibacillus* spp. altering energy allocation from fat storage to reproduction." This statement appears a little bit out of nowhere and is not mentioned enough before to claim "bacteria and food availability governs fat utilization".

Response: We have revised this section to improve the logical flow and provide clearer context.

- Methods: Line 412 contigs greater than 3kbp are kept, but line 422 MAGs longer than 1kbp are kept. I guess it is 1Mbps? In this case, it contradicts line 109, which states that 1.5 Mbps MAGs were kept.

Response: We thank the reviewer for pointing out the inconsistency. We have updated our method for filtering MAGs. Instead of using length as a primary filter, we filter by removing those without matched BUSCO housekeeping genes. This adjustment excluded 2 and 17 MAGs in the early and late stage, respectively, characterized by relatively low frequency. This revised approach resulted in all final MAGs exceeding the size of 1.5 Mb. These corrections have been duly noted in the Methods section of our manuscript.

- Legends: Bot instead of dot? (lines 619 and 666)

Response: We have corrected 'bot' to 'dot' in lines 619 and 666.

- Figure 1 B. Maybe adjusting the color in the stacked bar plot to taxonomy could help grasp significant differences between time points. (For example, the Gammaproteobacteria could be different shades of red).

Response: We thank the reviewer for the suggestion to enhance the visual representation in Fig. 1b. We understand the benefit of color-coding the stacked bar plot according to taxonomic classification. However, in our dataset, the MAGs are predominantly composed of Betaproteobacteria and Gammaproteobacteria, with the early stage alone featuring 24 and 27 MAGs, respectively. Additionally, when categorized at the order level, Burkholderiales and Enterobacterales represent significant portions with 24 and 13 MAGs, respectively. Given this composition, assigning distinct colors based on taxonomy could lead to a visually cluttered and potentially confusing plot due to the high number of similar entities.

- Figure 2. I really like this figure. It is a nice way to convey abundance and functionality. The ranking panel needs to be clarified. As I read from left to right, I assume 1st on the left, but it could be different for people reading from right to left. Adding a 1st/10th/20th on the x-axis could help read the figure.

Response: We thank the reviewer for the positive feedback on Fig. 2 and the suggestion. We have now included markers indicating the positions on the x-axis.

REVIEWER COMMENTS

Reviewer #1 (Remarks to the Author):

The authors' revisions and the supplementary experimental data have effectively addressed my concerns.

Reviewer #2 (Remarks to the Author):

The manuscript titled: "Microbiota Succession Influences Nematode Physiology in a Beetle Microcosm Ecosystem" has been thoroughly revised by the authors. I am satisfied with the authors' overall response to the review. I still have a few minor comments:

I am bothered by the new RNA experiment. The main reason is that different species of beetles are used. Although they belong to the same family but not genus, the two beetles are still sampled from different continents and might have some base nutrient composition differences, which will also impact the associated microbiome composition. I understand the change in the main authors' scientific institution and the subsequent change in available host sources. Nevertheless, the new experiment should be better integrated into the current version of the manuscript.

Using a second beetle species is only mentioned in the Material and Methods, and it feels hidden. It should be directly brought up in the results section. On line 230, "Additional comparisons" should be expanded to provide more details on the new comparison performed. Although the comparison is quite different from the original, I think there is some value in mentioning it early, as the author can discuss the fact that a lot of the Kegg-annotated gene expression is similar across different microcosms. It is an important factor that probably affects the large number of differentially expressed genes observed in the stage-specific dataset, and this should also be discussed. Overall, I think it could be an argument supporting the author's statement when they say they "created a laboratory microcosm to investigate the dynamics and succession of bacterial composition and the corresponding *P. pacificus* physiology."

To this end, comparing the 16S rRNA dataset of the nematode gut microbiome composition of both species could also be interesting.

Comments on the authors' answers:

we updated to HUMAnN 3.8, utilizing the most recent database. Nevertheless, we still encountered about 30% and 50% of raw reads from early and late stage, respectively, that could not be assigned (i.e. UNMAPPED and UNINTEGRATED). This is to be expected; using HUMAnN, you use a small reference database of known genes that is only a fraction of what microbial genomes comprise.

Several abundant genera, e.g. *Myroides*, *Lysinibacillus*, and *Pseudochrobactrum* were not detected by HUMAnN. This is strange; I know *Lysinibacillus* is part of the HUMANN database. Any idea as to why there is a discrepancy between the two approaches?

In the interest of maintaining consistency in our results, we ultimately decided to rely on the findings from our de novo assembly. Fair enough

We performed a cross-mapping of raw reads across different time points, and less than 3% of the reads could be mapped to MAGs predicted at alternate time points. This is interesting; it suggests that the communities are very different across time points. Even if the microbial association with the nematode is transient, I would still expect some dominant bacterial species to stay the same across time. I think here the metric to look at is not how many reads map to other MAGs but the individual MAG coverage. Even with centrifugation, your raw reads will include host DNA, which will reduce the overall raw read coverage. Except if you filtered the host from the dataset but I might have missed that from your material and methods.

We acknowledge the importance of identifying virulence genes, particularly those associated with nematode pathogenicity. I apologize for the lack of clarity in my previous comment. I meant performing a full annotation of the MAGs using a tool made to annotate bacterial genomes such as Bakta. This will provide better and faster annotation than blasting CDS against nr. Comparing full

annotations would be quite interesting, especially for bacteria from the same species. For example, for *Myroides odoratus*, you have four mags, one abundant in every sample, but three have a very different ranking compared to the control (and the first one). Full gene content could help identify what is important for "colonization" on the bacterial side." This being said the *Myroides* MAGs have a BUSCO score of around 50%, indicating they are quite incomplete. Still, you have cases like the "*Sphingobacterium* spp" with similar differences in ranking between high BUSCO score mags, which could yield informative comparisons.

We apologize for any confusion caused by the original presentation in Fig. 4, which depicted the absolute values of fold changes, combining both upregulated and downregulated genes. We have modified Fig. 4 to separate the genes that are upregulated (left) from those that are downregulated (right), and add gene counts for each category. Our analysis, supported by our GSEA results, which relies on different statistical approaches, reveals that there are more genes significantly downregulated on grubs. I am still confused by this explication and see no figure with split Up- and Down- regulated panels. If this is refers to Fig 3.a, an explication for the negative score of the panel is needed to improve clarity.

The manuscript titled: "Microbiota Succession Influences Nematode Physiology in a Beetle Microcosm Ecosystem" has been thoroughly revised by the authors. I am satisfied with the authors' overall response to the review. I still have a few minor comments:

I am bothered by the new RNA experiment. The main reason is that different species of beetles are used. Although they belong to the same family but not genus, the two beetles are still sampled from different continents and might have some base nutrient composition differences, which will also impact the associated microbiome composition. I understand the change in the main authors' scientific institution and the subsequent change in available host sources. Nevertheless, the new experiment should be better integrated into the current version of the manuscript. Using a second beetle species is only mentioned in the Material and Methods, and it feels hidden. It should be directly brought up in the results section.

Response: We gratefully appreciate the reviewer's comments, which have significantly improved the quality of this manuscript.

We thank the reviewer for understanding our situation and the rationale for using a different beetle. We apologize if the presentation of additional experiments with the second beetle made the reviewer feel that it was hidden, and we also apologize for not specifying this issue other than in the Materials and Methods. We have now clarified the use of the second beetle in the introduction and discussed the data in the results and discussion section.

On line 230, "Additional comparisons" should be expanded to provide more details on the new comparison performed. Although the comparison is quite different from the original, I think there is some value in mentioning it early, as the author can discuss the fact that a lot of the Kegg-annotated gene expression is similar across different microcosms. It is an important factor that probably affects the large number of differentially expressed genes observed in the stage-specific dataset, and this should also be discussed. Overall, I think it could be an argument supporting the author's statement when they say they "created a laboratory microcosm to investigate the dynamics and succession of bacterial composition and the corresponding *P. pacificus* physiology."

Response: We thank the reviewer's suggestions for the RNAseq-related analyses. We have improved the overall structure to make this part easier for readers to understand. Additionally, we have discussed the similarity of microbiota from different grubs and corresponding physiological changes in the nematodes.

To this end, comparing the 16S rRNA dataset of the nematode gut microbiome composition of both species could also be interesting.

Response: In response to the suggestion of comparing the 16S rRNA dataset for the gut microbiome composition of both species, our original approach involved analyzing the microbiota of the rose chafer through metagenomic sequencing. We have not generated corresponding results for the nematode gut microbiome, because such analysis would predominantly yield nematode DNA.

Currently, we lack access to grubs of the rose chafer, preventing us from conducting additional 16S rDNA analyses in this study. Nevertheless, we remain keenly interested in exploring the nematode gut microbiota and plan to undertake a systematic study on this topic in the near future.

Comments on the authors' answers:

we updated to HUMAnN 3.8, utilizing the most recent database. Nevertheless, we still encountered about 30% and 50% of raw reads from early and late stage, respectively, that could not be assigned (i.e. UNMAPPED and UNINTEGRATED). This is to be expected; using HUMAnN, you use a small reference database of known genes that is only a fraction of what microbial genomes comprise.

Several abundant genera, e.g. *Myroides*, *Lysinibacillus*, and *Pseudochrobactrum* were not detected by HUMAnN.

This is strange; I know *Lysinibacillus* is part of the HUMANN database. Any idea as to why there is a discrepancy between the two approaches?

Response: The HUMANN database includes sequences of *Myroides odoratus* and *Lysinibacillus sphaericus*, species assigned by our taxonomic classifications. The Metaphlan incorporated within the same HUMANN run have identified both *M. odoratus* and *L. sphaericus*, with ratios resembling those in our *de novo* approach. The reason why HUMANN failed to detect *M. odoratus* and *L. sphaericus* remains unclear to us.

In the interest of maintaining consistency in our results, we ultimately decided to rely on the findings from our *de novo* assembly.

Fair enough

We performed a cross-mapping of raw reads across different time points, and less than 3% of the reads could be mapped to MAGs predicted at alternate time points.

This is interesting; it suggests that the communities are very different across time points. Even if the microbial association with the nematode is transient, I would still expect some dominant bacterial species to stay the same across time. I think here the metric to look at is not how many reads map to other MAGs but the individual MAG coverage. Even with centrifugation, your raw reads will include host DNA, which will reduce the overall raw read coverage. Except if you filtered the host from the dataset but I might have missed that from your material and methods.

Response: We express our sincere appreciation for the reviewer's careful review of this matter. After a re-examination of our findings, we acknowledge that our previous interpretation of the results was partially incorrect. Specifically, we would like to clarify the above issue regarding the 3% mapping rate, which counts for all raw reads, including those not mapped to any MAGs. However, in our analyses of abundance, we only take into account the reads that have successfully mapped to MAGs. Consequently, this rate should not be used for drawing conclusions. At the later stage, where the number of MAGs increased, the proportion of most shared taxonomic species accounted for less than 0.5% of the quantity. Upon detailed analyses of specific MAGs, we observed substantial coverage in the nematode-free treatment, while the coverage for most of the nematode treatments remained low.

In our previous analysis, we considered MAGs with amino acid substitutions as different MAGs. However, further analyses of some of these MAGs with identical taxonomic classifications revealed that differences in gene presence and absence among the MAGs were less than 5%, which might be caused by the assembly process. Thus, despite the presence of some level of amino acid substitutions (with similarities > 99%) and nucleotide changes (~ 98%), these MAGs should be considered as the same strains.

This finding suggests that the sequence diversity observed might have been present in the initial bacterial populations. Differences in the ratios of single nucleotide polymorphisms (SNPs) between the early and late stage could explain the SNPs in the MAGs. This hypothesis was further supported by assembling raw reads from control groups at both early and late stages. For instance, in the case of *Serratia marcescens*, this process yielded two MAGs, not four, with an average nucleotide similarity of 99% from the original assembly. Therefore, these MAGs should be considered as the same strains and there are MAGs shared between stages. We have updated our manuscript to reflect these analyses and findings.

We have incorporated a detailed description in the Materials and Methods section, and annotated the MAGs identified at different stages which could be the same ones (Supplementary Table 1).

Additionally, we have determined the proportion of raw reads derived from *P. pacificus*. Among all the raw reads, those originating from the host accounted for approximately 0.7% and 5.1% at the early and late stages, respectively. Together, we feel that this careful re-analysis helps clarify the remaining issues from the reviewer. We again, thank the reviewer for this careful consideration.

We acknowledge the importance of identifying virulence genes, particularly those associated with nematode pathogenicity.

I apologize for the lack of clarity in my previous comment. I meant performing a full annotation of the MAGs using a tool made to annotate bacterial genomes such as Bakta. This will provide better and faster annotation than blasting CDS against nr. Comparing full annotations would be quite interesting, especially for bacteria from the same species. For example, for *Myroides odoratus*, you have four mags, one abundant in every sample, but three have a very different ranking compared to the control (and the first one). Full gene content could help identify what is important for "colonization" on the bacterial side." This being said the *Myroides* MAGs have a BUSCO score of around 50%, indicating they are quite incomplete. Still, you have cases like the "*Sphingobacterium* spp" with similar differences in ranking between high BUSCO score mags, which could yield informative comparisons.

Response: We appreciate the additional guidance provided. In response to the reviewer's feedback, we selected MAGs that appeared at both early and late stages for comparative genomic analyses. We have also performed annotations using Bakta and uploaded the gbff files to Zenodo.

We wish to address a likely underestimation of BUSCO scores for some MAGs, which may be attributed to the absence of ribosomal protein genes in the assembly phase. For instance, the genome analysis of *M. odoratus*, compared with two complete genome sequences in the NCBI genome database, indicates a gene count of approximately 3,600 within a genome size nearing 4.2 Mb, consistent with our findings for *M. odoratus* assemblies.

It is important to note that the majority of these bacterial species are not pathogenic or merely opportunistic pathogens. We did not identify known virulence/colonization genes, except for *S. marcescens*, where we could more clearly identify virulence-related genes. We have incorporated the results of our comparative genomic analysis into the manuscript, with the updated results presented in Supplementary Fig. 3c.

We apologize for any confusion caused by the original presentation in Fig. 4, which depicted the absolute values of fold changes, combining both upregulated and downregulated genes. We have modified Fig. 4 to separate the genes that are upregulated (left) from those that are downregulated (right), and add gene counts for each category. Our analysis, supported by our GSEA results, which relies on different statistical approaches, reveals that there are more genes significantly downregulated on grubs.

I am still confused by this explication and see no figure with split Up- and Down-regulated panels. If this is referes to Fig 3.a, an explication for the negative score of the panel is needed to improve clarity.

Response: We apologize for any confusion that remained. We have further refined our figures for clarity. Specifically, in Figure 4a, we present the fold change and

counts of both up- and down-regulated genes at both early and late stages. There is a higher number of down-regulated genes when *P. pacificus* is grown on decomposed grub, which contributes to the observed enrichment of down-regulated KEGG pathways. We have updated the results section accordingly and included the revised figure (Figure 4; shown below) to enhance our communication.

REVIEWERS' COMMENTS

Reviewer #2 (Remarks to the Author):

I am satisfied with the authors' new version of edits on the manuscript and recommend the manuscript for publication.